Species turnover reveals hidden effects of decreasing nitrogen deposition in mountain hay meadows

http://orcid.org/0000-0001-9742-8297 Roth Tobias 1 2 t.roth@unibas.ch
Kohli Lukas 2
Bühler Christoph 2
Rihm Beat 3
http://orcid.org/0000-0003-0059-3224 Meuli Reto Giulio 4
Meier Reto 5
Amrhein Valentin 1
1 Zoological Institute, University of Basel , Basel , Switzerland
2 Hintermann & Weber AG , Reinach , Switzerland
3 Meteotest , Bern , Switzerland
4 Swiss Soil Monitoring Network NABO, Agroscope , Zurich , Switzerland
5 Air Pollution Control and Chemicals Division, Federal Office for the Environment , Bern , Switzerland
Burns Douglas
Electronic publication date: 2019 Feb 6
Publication date: 2019
Volume: 7
Electronic Location ID: e6347
Received 2018 Sep 14; Accepted 2018 Dec 24
Copyright: © 2019 Roth et al.
Copyright year: 2019
Copyright holder: Roth et al.
License: This is an open access article distributed under the terms of the Creative Commons Attribution License, which permits unrestricted use, distribution, reproduction and adaptation in any medium and for any purpose provided that it is properly attributed. For attribution, the original author(s), title, publication source (PeerJ) and either DOI or URL of the article must be cited.
License URL: https://creativecommons.org/licenses/by/4.0/

Keywords: Alpine meadows, Nitrogen critical loads, Plant community composition, Species richness, Grassland, Biodiversity, Mountain hay meadows, Nitrogen deposition, Biodiversity monitoring

Funding: The FOEN, the Swiss National Science Foundation 31003A_156294 The Swiss Association Pro Petite Camargue Alsacienne, the Fondation de bienfaisance Jeanne Lovioz, and the MAVA Foundation This work was supported by the FOEN, the Swiss National Science Foundation (grant no. 31003A_156294), the Swiss Association Pro Petite Camargue Alsacienne, the Fondation de bienfaisance Jeanne Lovioz, and the MAVA Foundation. There was no additional external funding received for this study. The funders had no role in study design, data collection and analysis, decision to publish, or preparation of the manuscript.

==============================
Nitrogen (N) deposition is a major threat to biodiversity in many habitats. The recent introduction of cleaner technologies in Switzerland has led to a reduction in the emissions of nitrogen oxides, with a consequent decrease in N deposition. We examined different drivers of plant community change, that is, N deposition, climate warming, and land-use change, in Swiss mountain hay meadows, using data from the Swiss biodiversity monitoring program. We compared indicator values of species that disappeared from or colonized a site (species turnover) with the indicator values of randomly chosen species from the same site. While oligotrophic plant species were more likely to colonize, compared to random expectation, we found only weak shifts in plant community composition. In particular, the average nutrient value of plant communities remained stable over time (2003–2017). We found the largest deviations from random expectation in the nutrient values of colonizing species, suggesting that N deposition or other factors that change the nutrient content of soils were important drivers of the species composition change over the last 15 years in Swiss mountain hay meadows. In addition, we observed an overall replacement of species with lower indicator values for temperature with species with higher values. Apparently, the community effects of the replacement of eutrophic species with oligotrophic species was outweighed by climate warming. Our results add to the increasing evidence that plant communities in changing environments may be relatively stable regarding average species richness or average indicator values, but that this apparent stability is often accompanied by a marked turnover of species.

Introduction

Nitrogen (N) deposition is the entry of reactive nitrogen compounds into soil, water, and vegetation, input from the atmosphere to the biosphere. Since nitrogen is an essential plant nutrient and many species-rich communities are adapted to conditions of low nitrogen availability (Vitousek et al., 1997), the addition of nitrogen is likely to change these communities. Indeed, together with land-use and climate change, N deposition is one of the major threats to biodiversity (Sala et al., 2000; Bobbink et al., 2010; Murphy & Romanuk, 2013). While there is strong evidence for the reduction in diversity of species-rich grasslands due to increased N deposition (Stevens et al., 2004; Duprè et al., 2010; Maskell et al., 2010; Wesche et al., 2012), mountain grasslands have received less attention (Humbert et al., 2016).

In many parts of Europe, measures to reduce atmospheric pollution have successfully reduced emissions of nitrogen oxides since the late 1980s, with an according decrease in N deposition (Tørseth et al., 2012; Fowler et al., 2007). However, there are a number of factors that may prevent the recovery of plant communities that suffered from increased N deposition. Among others, N deposition is still high at many sites, since in contrast to nitrogen oxides, ammonia emissions decreased only to a small degree. Thus, even if N deposition is reduced, large areas might still be above the critical threshold above which harmful effects on plant diversity do occur (Bobbink et al., 2010; Slootweg, Posch & Hettelingh, 2015; Rihm & Achermann, 2016). Furthermore, there is a possibility that communities reach an alternative stable state after decades of increased N deposition and that the respective plant species are unlikely to disappear even if N deposition is reduced (Stevens, 2016). Alternatively, if oligotrophic species disappeared from the entire landscape, dispersal limitation may prevent oligotrophic species from recolonizing sites (Dirnböck & Dullinger, 2004). It is therefore an open question whether and how fast the reduction in N deposition rates will lead to the recovery of plant communities.

Recovery of existing plant communities after high N deposition would imply that the state of communities measured at different points in time is improving over time (i.e., improving biodiversity endpoints sensu Rowe et al. (2017)). Species richness, a biodiversity endpoint that can be relatively easily assessed and communicated, is often negatively related to nitrogen deposition (Maskell et al., 2010; Field et al., 2014; Rowe et al., 2017). Other metrics that are potentially more useful to reflect favorable changes can be derived from the traits of the species in a community. In Europe, environmental preference of plants has often been expressed using indicator values assigned to each plant species (Ellenberg et al., 1992; Landolt et al., 2010). Examples of such metrics would be the number of oligotrophic species, or the average indicator value of the species in a community (Roth et al., 2013; Rowe et al., 2017). However, the lack of a temporal trend in such biodiversity endpoints—particularly in species richness—does not necessarily mean that species composition remains unchanged. This is because immigration and extinction might be equally frequent and may cancel each other out (Hillebrand et al., 2018). Thus, a useful approach to understanding biodiversity change is through estimates of species turnover reflecting both colonization and local extinction (Hillebrand et al., 2018), especially if colonization and local extinction are compared to random expectation (Chase & Myers, 2011).

In Switzerland, grassland accounts for 70% of the agricultural land. With extensive cultivation, permanent grassland has a very high biodiversity. This applies in particular to the meadows in the alpine region, where meadows with high plant diversity are also of agronomical importance (Leiber et al., 2006). In mountain hay meadows, the spatial variation of species richness in vascular plants has been shown to be negatively correlated with N deposition (Roth et al., 2013), suggesting that mountain grasslands are negatively affected by increased N deposition. However, between 1990 and 2010, NOx emissions in Switzerland decreased by 46% and NH3 emissions by 14%, and considerable emission reductions also occurred in neighboring countries (Maas & Grennfelt, 2016). Potentially, this could have led to a partial recovery of plant communities.

In addition to N deposition, Swiss mountain ecosystems are also threatened by other drivers of global change. In Switzerland, temperatures increased from 1959 to 2008 at all altitudes, with an average warming rate of 0.35 °C per decade, which is about 1.6 times the northern hemispheric warming rate (Ceppi et al., 2012). Climate warming is likely to interact with N deposition in driving plant community changes (Humbert et al., 2016). Indeed, in an earlier study we found that, at the landscape scale, plant communities responded to climate warming within a relatively short period of time (Roth, Plattner & Amrhein, 2014). Steinbauer et al. (2018) suggest that particularly the shift of plant communities at mountain summits is the result of recent climate warming, and they assume an interaction with airborne N deposition. Furthermore, traditional management regimes are currently changing, which also has major impacts on plant communities in mountainous regions of Europe (Niedrist et al., 2009; Homburger & Hofer, 2012). Management regimes of easily accessible mountainous areas are often being intensified, while poorly accessible mountainous areas are abandoned (Tasser & Tappeiner, 2002; Strebel & Bühler, 2015). Note that fires—an important driver of biodiversity in other grassland communities (Ratajczak et al., 2014)—hardly occur in Central European mountain hay meadows.

Here, we used data from the Swiss biodiversity monitoring (BDM) program (Weber, Hintermann & Zangger, 2004) to address the following questions: (1) Did biodiversity endpoints that are likely to reflect temperature, precipitation, N deposition, or land-use intensity change over the last 15 years? (2) Was species turnover correlated with the average temperature, precipitation, N deposition, or inclination (we expect steep areas to be less intensively managed)? (3) Did species that newly colonized or disappeared from local sites differ from random expectation, according to their indicator values for temperature, soil moisture, nutrients, or light?

Materials and Methods

Monitoring data and community measures

We analyzed the presence/absence of vascular plants sampled in the Swiss BDM program that was launched in 2001 to monitor Switzerland’s biodiversity and to comply with the Convention on Biological Diversity of Rio de Janeiro (Weber, Hintermann & Zangger, 2004). The sampling sites were circles with a size of 10 m2, and data collection was carried out by qualified botanists who visited each sampling site twice within the same season. During each visit, all vascular plant species detected on the plot were recorded except for young plants that have not yet developed at least the first pair of leaves after the cotyledons. For details on the field methods see Plattner, Birrer & Weber (2004), Roth et al. (2013), and Roth et al. (2017).

After the sampling of the plant data, the botanists also assigned a habitat type to each sampling site according to the classification system developed for Switzerland (Delarze & Gonseth, 2008). We matched the habitat types of the Swiss classification system with the categories from the EUNIS system (level-3 classification; Davies, Moss & Hill, 2004) and selected all sampling sites in mountain hay meadows (EUNIS E2.3). We analyzed the data from 2003 to 2017. During that study period, each sampling site was surveyed once per 5-year period: the first period lasted from 2003 to 2007, the second from 2008 to 2012, and the third from 2013 to 2017. These selection criteria yielded 129 sites that had been sampled three times during 2003–2017. The distribution of sites across Switzerland is given in Fig. 1.

Figure 1 Distribution of the 129 study sites across Switzerland.

Background data source: Swisstopo, Federal Office of Topography.

Before the analyses we removed all records that were not identified on the species level. For each survey (that consisted of two visits per season) we then calculated the following biodiversity endpoints: We used the number of recorded species (species richness) that can be easily related to many conservation targets (Rowe et al., 2017). Additionally, we calculated the community mean (CM) of the Landolt indicator values of recorded species. Similar to the Ellenberg indicator values (Ellenberg, 1974), the Landolt values are ordinal numbers that express the realized ecological optima of plants species for different climatic, soil or land-use variables. The Landolt indicator values were developed for the specific situation in Switzerland, published the first time in Landolt (1977) and recalibrated in Landolt et al. (2010). Their predictive power was tested in different studies (Scherrer & Körner, 2011). We analyzed the indicator values for temperature (1: high elevation species; 5: low elevation species), soil moisture (1: species that grow in soils with low moisture; 5: species that grow in water-saturated soils), nutrients (referring in particular to nitrogen, but also to phosphorus; 1: species that grow in nutrient-poor soils; 5: species that grow under nutrient-rich conditions) and light (1: species that grow in shade; 5: species that predominantly occur in bright places).

In addition to the five biodiversity endpoints that describe the state of plant communities for each site at a given time point, we also estimated the temporal turnover (i.e., species exchange ratio sensu Hillebrand et al. (2018)) as the proportion of species that differ between two time points, to describe the community change between two points in time.

To test data quality in the BDM program, independent replicate surveys were routinely performed by botanists who were not involved in the regular BDM surveys. The regular surveyors did not know whether and in which sites their surveys were replicated (Plattner, Birrer & Weber, 2004). We used the data from 14 replicated surveys to calculate the pseudo-turnover, which is the proportion of species that differed between two surveys that were conducted by two different surveyors during the same year on the same site.

Environmental gradients

We expected different drivers of global change to cause temporal change in mountain hay communities. To better disentangle the importance of these mechanisms, we ordered the sites along four main environmental gradients. First, we expected communities to respond to climate warming (Roth, Plattner & Amrhein, 2014). To describe the temperature gradient, we used the mean annual temperature per site based on data from the WorldClim database (Fick & Hijmans, 2017). The average ± SD mean annual temperature at our sites was 5.85 ± 2.16 °C. The monthly-mean surface air temperature for Switzerland shows a linear increase of 1.29 °C per 100 years between 1864 and 2016 with the warmest three years of the entire period measured in 2011, 2014, and 2015 (Begert & Frei, 2018). Also based on data from the WorldClim database, we used the annual precipitation per site, another key driver for plants that is likely to be affected by climate change (Beier et al., 2012). The average annual precipitation at our sites was 1,284.71 ± 196.53 mm. Further, we estimated atmospheric N deposition for each site using a pragmatic approach that combined monitoring data, spatial interpolation methods, emission inventories, statistical dispersion models, and inferential deposition models (Rihm & Achermann, 2016). The average nitrogen deposition at our sites was 17.54 ± 6.47 kg ha−1 year−1 in 2000 and 14.84 ± 6.12 kg ha−1 year−1 in 2015. We assume that N deposition is a surrogate for N availability in the soil because we found that the spatial variation in oligotrophic species richness is clearly linked to N deposition (Roth et al., 2013, 2017); unfortunately, we do not have soil measurements to test this assumption. Apparently, the total N as well as the soil carbon content down to 20 cm depth were mostly stable over the last 20 years in the extensively used grassland sites of the Swiss soil monitoring network (NABO; R. Meuli, 2018, personal communication). Finally, we used inclination as a proxy for land-use intensity, because we assumed that steeper sites are likely to be less intensively managed (Strebel & Bühler, 2015). The average inclination at our sites was 15.87 ± 9.66°.

Statistical analyses

To estimate the linear trend over time for each of the five biodiversity endpoints, we applied linear mixed models (LMMs) with normal distribution except for species-richness with Poisson distribution. We specified site-specific trends with the assumption that the between-site differences in intercepts and slopes can be described with normal distributions (i.e., a random intercept random slope model, Gelman & Hill, 2006). Model parameters were estimated in a Bayesian framework using the R-Package rstanarm (Stan Development Team, 2016; Muth, Oravecz & Gabry, 2018).

To infer whether species turnover was changing along the gradient, we used a binomial generalized linear mixed model (GLMM) with the proportion of species that differed between two surveys as dependent variable, with the site gradients, the period (first/second vs second/third surveys) and the number of recorded species as predictors, and with site-ID as random effect. Model parameters were estimated in a Bayesian framework using the R-Package rstanarm (Stan Development Team, 2016; Muth, Oravecz & Gabry, 2018).

To infer whether species that colonized a site or disappeared from a site had particular indicator values that differed from the other species at that site, we produced for each site a list with all species that were recorded during the three surveys (total community). We then calculated the CM of the indicator value for all species that colonized the site during the three surveys (i.e., not recorded during the first survey and recorded during the second, or not recorded during the second and recorded during the third survey). We then randomly selected the same number of species from the total community and also calculated the CM of the value for these species (random-CM). We repeated the random selection of species 1,000 times. We then calculated the differences of the CM minus the average of the random-CMs to obtain a standardized difference (standardized-CM) of how different the colonizing species were from random expectation. For example, a difference <0 would suggest that the indicator values of colonizing species were lower than might be expected from random colonization from the species-pool for this site. We applied this method for both colonizing and disappearing species and for the indicator values for temperature, soil moisture, nutrients, and light (see Appendix A). We then tested whether this standardized difference was changing along the corresponding gradient (inferred from independent datasets, see Environmental gradients section) using linear models. Model parameters were estimated in a Bayesian framework using the R-Package rstanarm (Stan Development Team, 2016; Muth, Oravecz & Gabry, 2018).

We used logistic GLMMs to analyse whether the colonization probability or local survival probability was changing along the nitrogen deposition gradient and whether this change depended on the species indicator value for nutrients. For the analysis of the colonization probability we examined for all species that were not observed at the first or second survey whether they were observed in the subsequent survey (first vs second, second vs third). Whether they were observed in the subsequent survey was used as dependent variable in the logistic GLMM. As predictor variables we used the N deposition of the site, the average indicator value for nutrients, and the interaction of these two variables. Additionally, species-ID and site-ID were included as random effects. The same logistic GLMM was also used to investigate local survival probabilities. In that case, however, we selected all species that were recorded during the first or second survey, and inferred whether or not the species was observed during the subsequent survey (first vs second, second vs third). Model parameters were estimated in a Bayesian framework using the R-package arm (Gelman & Su, 2018).

To estimate the effect of N deposition on total species richness at a given point in time, we described the plant species richness at the sites using a generalized linear model (GLM) with Poisson distribution and the logarithm as link function. As predictors we used the four environmental gradients as described above. Model parameters were estimated in a Bayesian framework using the R-package arm (Gelman & Su, 2018).

As parameter estimates we give the 5% and 95% quantiles of the marginal posterior distribution, which we interpreted as a 90% compatibility interval showing effect sizes most compatible with the data, under the model and prior distribution used to compute the interval (Amrhein, Trafimow & Greenland, in press). Smoothing was done using the “loess” function of the R-library “stats” (R Development Core Team, 2018) with default settings.

Data accessibility and reproducibility of results

Data and R Markdown documents (Manuscript.Rmd and Appendix_A.Rmd) to fully reproduce this manuscript including figures and tables are provided at https://github.com/TobiasRoth/NDep-Trend. An R Markdown document is written in markdown (plain text format) and contains chunks of embedded R code to produce the figures and tables (Xie, Allaire & Grolemund, 2018). Raw data for analyses are provided in the folder “RData” and the folder “R” contains the R-Script that was used to export the data from the BDM database. The folder “Settings” contains a list of all the R packages (including version number) that were in the workspace when the manuscript was rendered.

The v1 release of the GitHub repository is the version that corresponds to the initial submission (https://github.com/TobiasRoth/NDep-Trend/releases/tag/v1), v2 is the version of the repository that corresponds to the revised version of the manuscript. The final version of the repository was archived at Zenodo (https://zenodo.org/record/2542933).

Declaration of reporting decisions

This paper presents a selection of analyses with results that appeared most promising or interesting to us. Our study should therefore be understood as being exploratory and descriptive.

Results

Plant communities

In total, 623 plant species were recorded on the 129 plots. Including the data of all three visits, 45.83 ± 11.54 (average ± SD) species were observed per plot. The lowest number of species recorded during a survey was 19 species and the highest number was 81 species. On average, 7.70% ± 8.20% of the recorded species were annual species with one plot reaching up to 50.00% annual species. The average indicator value for temperature across all surveys was 3.12 ± 0.37, ranging from 1.59 to 3.66. The average indicator value for soil moisture was 2.99 ± 0.20, ranging from 2.46 to 3.59. The average indicator value for nutrients was 3.20 ± 0.35, ranging from 2.26 to 4.00. And the average indicator value for light was 3.55 ± 0.19, ranging from 2.83 to 4.16.

Temporal change in community structures

Species richness and the four measures of plant community structure according to Landolt indicator values (i.e., biodiversity endpoints) suggested that plant communities in mountain hay meadows were rather stable between 2003 and 2017 and did not show a clear increase or decrease over time (Table 1): for each of the three 5-year survey periods, the averages of species richness and the average indicator values for temperature, soil moisture, nutrients, and light did not vary much among the three sampling periods, and the estimated trends were rather small. The results from the LMMs suggest that a linear temporal change was most likely for the community mean (CM) of the indicator value for temperature (probability of increase: 0.97), followed by the CM of the indicator value for light (probability of decrease: 0.93). A linear temporal change was least likely for the species richness (probability of increase: 0.53). The probability that the CM of the nutrient value decreased between 2003 and 2017 was 0.67.

Table 1 Average measures of the biodiversity endpoints for the three sampling periods (period 1: 2003–2007; period 2: 2008–2012; period: 2013–2017).

Measures	Period 1	Period 2	Period 3	Trend	5%	95%	Probability for trend	
Species richness	45.72	46.02	45.74	0.00	−0.03	0.03	0.53	
Temperature value	3.11	3.13	3.13	0.01	0.00	0.03	0.97	
Soil moisture value	2.99	2.98	2.99	0.01	−0.01	0.02	0.80	
Nutrient value	3.20	3.20	3.20	0.00	−0.02	0.01	0.33	
Light value	3.56	3.55	3.55	−0.01	−0.02	0.00	0.07	
Note:

The temporal trends are given as change per 10 years and were estimated from linear mixed models with normal distribution (except for species richness with Poisson distribution and a log-link function). The measure of precision for the temporal trend is given as the 5% and 95% quantiles of the marginal posterior distribution of the linear trend (90% compatibility interval). The column “Probability for trend” gives the probability that the linear trend is >0. Indicator values according to Landolt et al. (2010).

Species turnover

The average ± SD percentage of species that differed between replicated surveys (i.e., different botanists surveyed the sites) was 28.81% ± 8.68%. This turnover between replicated surveys was lower than the observed temporal turnover: the average percentage of species that differed between the first and second survey at a site was 37.65% ± 10.43%, and the percentage of species that differed between the second and third survey was 35.66% ± 10.36%. Thus, it seemed that the turnover from the first/second survey to the turnover of the second/third survey moderately decreased (period effect in Table 2). Species richness was a good predictor of species turnover: species rich sites were subject to higher turnover than sites with lower species richness (Table 2). The four gradients (temperature, precipitation, nitrogen deposition, and inclination) were less conclusive in explaining the variation in species turnover among sites.

Table 2 Change of species turnover along the four gradients when differences between the two periods (first period: turnover between first and second survey; second period: change in turnover between second and third surveys and species richness effects are accounted for).

Predictors	Estimate	5%	95%	
Period	−0.09	−0.15	−0.03	
Number of species	0.14	0.09	0.18	
Mean annual temperature	0.04	−0.02	0.10	
Mean annual precipitation	−0.06	−0.15	0.03	
Nitrogen deposition	−0.07	−0.23	0.08	
Inclination	−0.07	−0.14	0.00	
Note:

Estimates for the period effect (change in turnover from first to second survey), the species richness effect (change in turnover per 10 species) and along the four gradients (slopes) with the corresponding 5% and 95% quantiles of the marginal posterior distribution were obtained from a binomial GLMM.

High species turnover at a site is the result of species that disappeared from the site and/or of species that newly colonized the site. To better understand the factors driving these changes we were particularly interested in whether the species that disappeared or colonized the sites differed in indicator values compared to what would be expected if the same number of species randomly disappeared or colonized the sites (i.e., random disappearance and random colonization), and whether there is spatial variation along the environmental gradients. It seems that the indicator values of newly colonizing species differed more from random colonization than the indicator values of disappearing species (Table 3). For colonizing species, we found the largest differences from random colonization in the indicator value for nutrients: at sites with relatively low N deposition of 10 kg ha−1 year−1, the newly colonizing species had on average a lower indicator value for nutrients than species under random colonization (column “Difference from random” in Table 3), but the differences between colonizing species and random colonization decreased with increasing N deposition (column “Change along gradient” in Table 3). Thus, at high N deposition, colonizing species did not differ from random species (see Appendix A).

Table 3 Difference in the average indicator value of species that (A) disappeared from a site or (B) newly colonized a site compared to the same number of species that were randomly selected from all species recorded at a site.

	Difference from random	Change along gradient	
Indicator value	Gradient	Estimate	5%	95%	Estimate	5%	95%	
(A) Plants that disappeared from a site	
Temperature	Annual mean temperature	−0.013	−0.035	0.006	0.007	−0.002	0.015	
Soil moisture	Annual mean precipitation	−0.002	−0.040	0.035	0.008	−0.014	0.030	
Nutrients	Nitrogen deposition	−0.019	−0.055	0.015	0.016	−0.027	0.059	
Light	Inclination	−0.022	−0.049	0.004	−0.002	−0.026	0.021	
(B) Plants that newly colonized a site	
Temperature	Annual mean temperature	0.017	0.002	0.033	−0.001	−0.008	0.006	
Soil moisture	Annual mean precipitation	0.021	−0.011	0.052	−0.002	−0.021	0.016	
Nutrients	Nitrogen deposition	−0.076	−0.106	−0.044	0.058	0.020	0.094	
Light	Inclination	−0.038	−0.061	−0.016	0.013	−0.008	0.032	
Note:

Shown are the results from linear models, with the difference between disappeared/colonized species and random species as dependent variable and the site-measure (gradient) as predictor variable.

While colonizing species had higher temperature values compared to what we would expect under random colonization, the effect size (i.e., the absolute value of the estimate for the difference from random expectation) for temperature was about four times smaller than the effect size for nutrients. Nevertheless, the variation in the indicator value for temperature seems to be important for explaining the total species turnover. This is because disappearing species tend to have lower temperature values than random species, and colonizing species tend to have higher temperature values than random species; both processes lead to an overall replacement of species with lower temperature value by species with higher temperature values. This was not the case for the indicator value for nutrients: species with lower nutrient values tended to be more likely to disappear from, and to colonize sites compared to random species (Table 3). See also Appendix A, where we present detailed results for the comparison between colonizing or disappearing species with randomly selected species.

Potential effects of reduction in nitrogen emissions

Nitrogen deposition decreased with increasing elevation (Fig. 2A). In 2000, only 11.63% of sites had a N deposition rate of less than 10 kg N ha−1 year−1, all of which situated above 1,000 m. Between 2000 and 2015, the N deposition decreased on average by −2.70 ± 1.74 kg N ha−1 year−1, with slightly higher net decreases at sites with previously high N deposition (Fig. 2B).

Figure 2 (A) The nitrogen (N) deposition in 2000 and (B) the change in N deposition between 2000 and 2015, along the N deposition gradient of the study sites used in 2000.

The blue lines indicate the loess curve and the grey areas indicate the corresponding 90% compatibility intervals.

In Fig. 3, we compare the colonization and local survival probability of oligotrophic species (indicator value of nutrients <3) and eutrophic species (indicator value of nutrients >3) along the N deposition gradient. Local survival probability was the same for oligotrophic and eutrophic species at a deposition rate of 12.66 kg N ha−1 year−1; colonization probability was the same for oligotrophic and eutrophic species at a deposition rate of 12.21 kg N ha−1 year−1. In 35.66% of the sites, the deposition rate was below 12.5 kg N ha−1 year−1 at which the replacement of eutrophic with oligotrophic species is likely, according to Fig. 3.

Figure 3 Colonization (A) and local survival (B) of oligotrophic species (indicator value for nutrients <3; red line) and of eutrophic species (indicator value for nutrients >3) along the N deposition gradient.

Given are means and 90% compatibility intervals from logistic linear mixed models. The vertical lines indicate the deposition rate with equal colonization or survival probabilities for oligotrophic and eutrophic species, the solid line indicating the median, and the dashed lines the 5% and 95% quantiles of the marginal posterior distribution.

While, we did not observe a consistent decrease in the average indicator values for nutrients (Table 1) the nutrient value of colonizing species was below average (i.e., a strongly negative effect size) at low deposition rate, and the difference to random expectation became smaller along the N deposition gradient (second last line in Table 3). This higher colonization rate of species with low nutrient value at sites with low N deposition rate might have affected the spatial pattern of oligotrophic species richness: sites with low N deposition were likely to become more species-rich over time. This likely resulted in a steeper slope of the negative relationship between N deposition and oligotrophic species richness, when comparing this relationship with a spatial approach as in Roth et al. (2013). Indeed, if we apply such an approach at different points in time to infer the effects of N deposition on the spatial variation of oligotrophic species richness, the resulting effect size (i.e., the slope) became more negative over time (Fig. 4).

Figure 4 Effects of N deposition on oligotrophic species richness estimated from applying the Poisson GLM, with species richness as dependent variable and N deposition plus other site covariates (elevation, precipitation, inclination, mean indicator values for soil moisture and light) as predictors, using only the surveys from one 5-year interval.

Note that within every 5-year interval, all plots were sampled once. Effect sizes are given as averages (points) and 5% and 95% quantiles (lines) of the marginal posterior distribution. The dashed blue line gives the linear regression with the effect size as dependent variable and the 5-year interval as predictor variable. The blue area gives the 5% and 95% quantiles of the marginal posterior distribution of the regression line.

Discussion

Although N deposition declined between 2000 and 2015 (Fig. 2), we observed only weak shifts in plant community structure (i.e., biodiversity endpoints sensu Rowe et al. (2017)) during the same time period (Table 1). While the slight increase in average temperature indicator values suggests that plant communities adopted to increasing temperatures, the constant average nutrient value suggests that the decrease in N deposition did not yet affect plant communities. However, this apparent stability in community composition was accompanied by a marked temporal turnover in species identities. It seems unlikely that this temporal turnover can entirely be explained by methodological issues such as overlooked species. First, pseudo-turnover of species entities in independent surveys of the same site during the same season was smaller than the observed temporal turnover between two surveys from different years. Second, spatial variation of turnover showed patterns that can hardly be explained by methodological issues. For instance, species turnover varied along the N deposition gradient, with highest species turnover at sites with low N deposition (Table 2). Taken together, our results add to the increasing evidence that contemporary plant communities may be relatively stable regarding average community composition, but that this apparent stability is often accompanied by a marked turnover of species (Vellend et al., 2013; Dornelas et al., 2014; Hillebrand et al., 2018).

Species communities are shaped by a range of factors, including deterministic processes such as environmental filtering or competitive interactions (Götzenberger et al., 2012; Janeček et al., 2013). Such factors select for species with specific characteristics. Community assembly theory thus suggests that the factors driving the composition of species in a community can be inferred from comparing the characteristics of the species in the community with random expectation (Chase & Myers, 2011). We adopted this idea and compared the indicator values of species that disappeared or colonized a site with indicator values from randomly chosen species from the same site (Appendix A). We found that the nutrient values of colonizing species showed the largest deviations from random expectation (Table 3), suggesting that in our sample, N deposition or other factors changing the nutrient content of soils were drivers of the change in species composition over the last 15 years.

In Swiss mountain hay meadows, the average N deposition was still rather high with an average of 14.84 kg ha−1 year−1, which is at the upper limit of the suggested critical load (Roth et al., 2017). Furthermore, nitrogen deposition only weakly decreased by about −2.70 kg ha−1 year−1 between 2000 and 2015. This is only about one tenth of the decrease in England, where N deposition decreased by 24 kg ha−1 year−1 from 1996 to 2011 (Storkey et al., 2015). The still comparatively high N deposition rate and the rather low decrease in N deposition, combined with the fact that most of the species are perennials, likely explain why we observed no change in average nutrient value of communities. Additionally, other anthropogenic pressures such as climate change might have outweighed effects of N deposition on community composition. In particular, we found that species disappearing from the sites tended to have below average indicator values for temperature, while species that newly colonize sites had above average indicator values for temperature (Table 3). Thus, the effect of disappearing and the effect of colonizing species on the community mean (CM) for temperature is additive, resulting in increasing average temperature values (Table 1). This was in contrast to how N deposition is affecting disappearance and colonization of species: It seems that both the species disappearing from the sites as well as species colonizing the sites tended to have below average indicator values for nutrients (Table 3). Thus, the effects of disappearing and colonizing species on the average community value for nutrients partially cancelled each other out. Furthermore, eutrophic species had rather high local survival across the entire deposition gradient, while oligotrophic species had much reduced local survival at higher N deposition rates. This suggests that mountain hay meadow communities can reach alternative stable states, with eutrophic species that are unlikely to disappear even if N deposition is reduced (Stevens, 2016). Taken together, these factors might explain why the composition of mountain hay meadow communities responded stronger to climate warming than to nitrogen reduction, although the reduction in nitrogen resulted in above average colonization of oligotrophic species.

Our results conform to the patterns described in recent reviews on biodiversity change, suggesting that local-scale species communities are often undergoing profound changes, but do not necessarily show a systematic loss of species numbers (Dornelas et al., 2014). However, our comparison with replicated surveys from the same year warn that an important portion of the observed turnover of species might be due to pseudo-turnover (i.e., species difference between two surveys that were conducted during the same year on the same site, but by two surveyors). Given that the BDM program has included major efforts in developing reproducible methods and has continuously invested in quality control (Plattner, Birrer & Weber, 2004), the recorded pseudo-turnover was quite high. A potential explanation is that species that are difficult to identify were only identified by one botanist at species level, while the other identified them at genus level. Although both botanists discovered the species this might have increased the pseudo-turnover because we only analyzed records that were identified at species level. Furthermore, replicated surveys are not conducted during the same days and in few cases the situation might have changed profoundly for example because the meadow was cut between the surveys. Such problems must be taken into account when evaluating the presented results. For example, we found that sites with high species richness had higher species turnover than sites with low species richness. This seems biologically plausible, since the average species coverage in species-rich sites must be lower than in species-poor sites, and species with low coverage probably have a higher turnover. At the same time, however, the result could also simply be due to pseudo-turnover, since species identification in species-rich sites is probably more difficult than in species-poor sites. However, for the presented result suggesting differences in colonization or local survival in relation to the species indicator values, we can hardly imagine how this could be caused by methodological issues.

Observational studies along a gradient of N deposition often conclude how the spatial variation in species richness is related to N deposition (Stevens et al., 2010b; Roth et al., 2013; chapter 4 in De Vries, Hettelingh & Posch, 2015). Such studies assume that the spatial variation in species richness (or other metrics of community composition) arose because of unequal species loss of different areas over time, resulting from elevated N deposition chronically experienced in some areas. Although there is evidence supporting the pertinency of such a “space for time substitution” for detecting the effects of N deposition on plant diversity (Stevens et al., 2010a), this approach cannot replace studies that relate temporal patterns in species composition with N deposition (De Schrijver et al., 2011). There are only a limited number of studies directly relating temporal trends of plant species diversity to varying amounts of N deposition in existing communities (Clark & Tilman, 2008; Storkey et al., 2015; Stevens, 2016). In an earlier study, we used the mountain hay meadow data from a single survey and estimated the empirical critical load along the N deposition gradient at which species richness of oligotrophic species richness starts to decrease with increasing N deposition (Roth et al., 2017). Using this spatial variation in species richness and N deposition, Roth et al. (2017) estimated a critical load for mountain hay meadows of 13.1 kg ha−1 year−1. In the current study, we estimated the rate of N deposition at which local survival probability or colonization probability was equal for oligotrophic and eutrophic species. Using a temporal approach in the present study, we obtained very similar results as Roth et al. (2017) using the spatial approach. Our results may thus be taken to validate the space for time approach, at least for Swiss mountain hay meadows.

However, Fig. 4 also shows that the results of spatial comparisons must be interpreted carefully. When we investigated the spatial variation in oligotrophic species richness with the same covariates as in Roth et al. (2017) for different study periods, to infer how oligotrophic species richness was decreasing along the N deposition gradient, the relationship appeared to vary between study periods. The decrease gradually became steeper (more negative) over time, except for the first two study periods. Our first interpretation was that the N deposition effect became stronger over time. This was against our prediction that the effect of N deposition should become weaker over time, since N deposition was decreasing during the study period. Then we realized that species turnover was highest at low N deposition sites (Table 2). At low N deposition rates, colonizing species have below average indicator values for nutrients. It seems that the decrease in N deposition resulted in oligotrophic species replacing eutrophic species particularly at sites with low N deposition. This seems to explain why the decline in oligotrophic species richness inferred from spatial patterns of species richness and N deposition is becoming steeper over time. And this may be interpreted as evidence that plant communities are recovering at least at low deposition sites and that in general the negative N deposition effects have not become stronger over time.

Conclusions

Comparing the indicator values of colonizing and disappearing species with random expectation, we found that oligotrophic species are currently more likely to colonize mountain hay meadows than eutrophic species, which might be the result of the recently observed decrease in atmospheric N deposition. However, our results also indicate that the recovering of mountain hay meadows from high N deposition might take much longer than transferring species-rich mountain hay meadows to species-poor communities with a large proportion of eutrophic species. This is because eutrophic species have high local survival probabilities, even after N deposition decreases again. Our study adds to the understanding of contemporary biodiversity change (Magurran et al., 2018), and it supports the notion of Hillebrand et al. (2018) that considering species turnover will generate a far more reliable view of the biotic response to changing environments than solely tracking community composition.

Supplemental Information

Supplemental Information 1 Appendix A: Comparison of lost/gained species to randomly selected species.

Click here for additional data file.

We thank the dedicated and qualified botanists who conducted fieldwork for the Swiss Biodiversity monitoring program. The Swiss Federal Office for the Environment (FOEN) kindly provided biodiversity monitoring data and topographic data.

Additional Information and Declarations

Competing Interests

Author Contributions

Data Availability

The authors declare that they have no competing interests.

Tobias Roth analyzed the data, prepared figures and/or tables, authored or reviewed drafts of the paper, approved the final draft.

Lukas Kohli analyzed the data, prepared figures and/or tables, authored or reviewed drafts of the paper, approved the final draft.

Christoph Bühler analyzed the data, authored or reviewed drafts of the paper, approved the final draft.

Beat Rihm authored or reviewed drafts of the paper, approved the final draft.

Reto Giulio Meuli authored or reviewed drafts of the paper, approved the final draft.

Reto Meier authored or reviewed drafts of the paper, approved the final draft.

Valentin Amrhein authored or reviewed drafts of the paper, approved the final draft.

The following information was supplied regarding data availability:

GitGub: https://github.com/TobiasRoth/NDep-Trend.

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
