# Peer review of "Species turnover reveals hidden effects of decreasing nitrogen deposition in mountain hay meadows"

_PeerJ, doi:10.7717/peerj.6347_

## Round 0.1 · original submission · Minor Revisions

The two reviewers believe your paper has merit and each had some constructive suggestions to improve the presentation and clarity. Please carefully consider the review comments and submit a revised manuscript with the next 40 days.

Reviewer 1 ·

Basic reporting

This study tests the hypothesis that decreasing nitrogen deposition over 14 years causes changes in plant community composition and rates of species turnover. Because the climate has also been warming and some sites experienced land use change, they also tested effects of temperature, precipitation, slope (inclination) and light in addition to nitrogen.

Experimental design

The researchers used plant species data from the Swiss Biodiversity Monitoring program and assigned them Ellenberg indicator values. The community mean indicator values were subjected to statistical linear mixed models to relate to environmental parameters and to assess species turnover rates.

Validity of the findings

Both declines in nitrogen and increases in temperature were important in driving changes in community composition over time, and somewhat counterbalanced each other.
Overall I found the approach solid and the conclusions well-supported by the data and analyses. I have some questions and suggestions for clarification of some points.

Additional comments

A total of 129 plots were selected based on their location in hay meadows. How far apart were the plots? It would be helpful to have a map with plot locations across the country, perhaps in the appendix.
Changes in N deposition over time are shown in Fig. 1. Since temperature was such an important driver, it would be useful to present changes in temperature over time. This could be reported in the text, if a figure is not of interest.

No information is presented on plant species composition. What are richness values per plot, and ranges of richness? No plant species names are given, and are not expected in such a diverse assemblage. However, to help the interested reader understand the species composition, the authors should refer to publication(s) that describe the vegetation and species composition of their sample sites.

The term “humidity” is used throughout to refer to the effects of precipitation on soil moisture. However, in English, humidity commonly refers to atmospheric humidity. I recommend substituting “soil moisture” or just “moisture” for humidity throughout the manuscript.

104 “community mean of indicator values”
It would be helpful to explain briefly how these indicator values are determined. Non-European readers may not be familiar with Ellenberg indicator values.

The results from plots with low N deposition are important to the conclusions and interpretations, but data for sites with low N deposition is not presented. In line 227 you state: “the higher colonization rate of species with low nutrient value at the few sites with low N deposition rate seems to affect the spatial pattern of oligotrophic species richness”
However, Fig. 3 only shows mean values for all sites. Is there a way you could show data from low N deposition sites, to support your statement on line 227?
Conclusions: 313 “where some key information is still missing”
What information is still missing?

Additional suggestions for editorial changes:

line 17 “with an according decrease in N deposition”
CHANGE TO
with a consequent decrease in N deposition

18 “Swiss biodiversity monitoring.”
Monitoring is an adjective in this usage and needs a noun to modify.
CHANGE TO
Swiss biodiversity monitoring program (as in l. 90)

22 “plant community structure”
CHANGE TO
“plant community composition” to be consistent with the rest of the manuscript.

Introduction
41 “with an according decrease in N deposition”
CHANGE TO
with a consequent decrease in N deposition

62 “if colonization and local extinction is compared”
CHANGE TO
if colonization and local extinction are compared

71 “are threatened also”
CHANGE TO
are also threatened

74 “likely to to interact”
CHANGE TO
likely to interact

77 “the shift of plant communities at mountain summits are”
CHANGE TO
the shift of plant communities at mountain summits is

82 “Swiss biodiversity monitoring”
CHANGE TO
Swiss biodiversity monitoring program

101 “These selection criteria yielded a sample of 129 sites.”
CHANGE TO
These selection criteria yielded 129 sites that had been sampled three times during 2003-2017 (is my interpretation correct?)

118 “we expected communities to response”
CHANGE TO
we expected communities to respond

194 “were less conclusive to explain the variation”
CHANGE TO
were less conclusive in explaining the variation

Table 3
(a) Plants that disappeard
CHANGE TO
(a) Plants that disappeared

The wording is unclear in this sentence:
208 “the differences between colonizing species and random colonization was about four times smaller than the difference in the indicator value for nutrients between colonizing species and random species.”
IS THIS INTERPRETATION CORRECT?
….the differences between RICHNESS OF colonizing species and random colonization was about four times smaller than the difference in the indicator value for nutrients between colonizing species and random species.

212” replacement of species with lower temperature value with species with higher temperature values.”
CHANGE TO
replacement of species with lower temperature value by species with higher temperature values.

227 “the higher colonization rate of species with low nutrient value at the few sites with low N deposition rate seems to affect the spatial pattern of oligotrophic species richness”
Data for sites with low N deposition is not presented. Fig. 3 shows mean values for all sites. Is there a way you could show data from low N deposition sites, to support your statement on line 227?

265” how N deposition is affecting disappearing and colonization”
CHANGE TO
how N deposition is affecting disappearance and colonization

Reviewer 2 ·

Basic reporting

The basic reporting is appropriate, only a few issues to mention:

Line 46-49: A third major factor is dispersal limitation. The authors should discuss either (a) why this is not an issue for the Swiss mountain hay meadows, or (b) include it in their assessment.

Figure 3 needs to be fixed. We can’t see the black line behind the confidence interval, and it’s not clear what this is telling us.

Experimental design

The experimental design and methodology are robust, but there were a few areas that could be clarified more and/or elaborated upon:

Line 105-106: There’s a lot that has to be taken on faith with respect to the indicator values from Landolt et al. (2010). This paper needs to stand on its own, and so these indicator values need to be summarized somehow and/or a table in the appendix provided on their values for different species. If it’s based on Ellenberg, I’ve seen many papers try to extract meaning from Ellenberg values in the past, most fail, and I wish the European ecologists would update that resource. That being said, this is one of the most successful efforts (if Landolt 2010 is based on Ellenberg 1992) I’ve seen in extracting meaning from those indicators, which is another testament to the strength of the analytical approach.

Line 114: Please provide a map of the sites and a table of their environmental values. That is critical background information that is lacking.

Validity of the findings

The validity of the findings is also solid, but could be tightened a little more in a few places:

Line 125-126: The average N deposition in the sites (14.84 kg N ha-1 yr-1) is still pretty high, and didn’t decline all that much in the 15 years, which also likely explains some of why you’re not seeing a large effect. It’ll be interesting to resample these sites in another 15 years, or to add more sites that have changed more. This should be discussed a little more.

Line 187: Interesting that you had 29% difference in the replicate plots assessed by different botanists, that seems quite big to me if the “signal” is 36-38% difference in the survey plots. There’s not much that can be done about this, and it’s very insightful to check, but some greater discussion on why the variation among replicate plots is so high is warranted.

Line 193: You found the “highest species turnover at sites with low N deposition.” That made me wonder, are these sites ones that used to have higher deposition and low are lower (and thus would be indicative of recovery) or sites that always had low N deposition (which would not be recovery). From Figure 1 it looks like these sites never had high N, so there may be something else driving these patterns other than N deposition? Some greater discussion of this possibility is needed.

Do you have any soil measurements that you can see if soil N mineralization or some over measure of soil N availability changed? Seems that is a key missing factor in the analysis. You’re assuming that N dep is a surrogate for N availability, which is may not be after decades of N deposition build up N stores in the soil. If that data is available, please include, if it isn’t, please state that.

It is a little unclear from the methods whether the “colonizers” needed to persist, or whether the species lost needed to not return? This is kind of important because otherwise you’re really just reporting on the seed rain (for colonizers), as colonizers need to persist to matter, and reporting on botanical skill (for those “lost”).

Figure 2: Very cool results!! Impressive that you were able to tease this apart. That being said, I would not call this a “critical load.” A critical load, according to the formal definition, is the level below which bad stuff doesn’t happen. I would not say that once the colonization probability of oligotrophic and eutrophic species are equal qualifies. This is a “threshold” to me. The critical load would be the point, if you had a saturating logarithmic response, where the colonization probability of eutrophic species began to increase, or where the colonization probability of oligotrophic species began to decrease. It’s not where they are equal, though where they are equal is a useful threshold to bring up.

Additional comments

Summary of comments
In this manuscript entitled “Species turnover reveals hidden effects of decreasing Nitrogen deposition in mountain hay meadows,” Roth et al. use a very insightful analytical approach to try and tease out the shift in plant community composition in Swiss mountain hay meadows over a 15 year period. They find that increasing temperature and decreasing nitrogen (N) deposition are having detectable effects on rates of colonization and local extirpation in the community, but they are subtle and often inconclusive. This is a nice (albeit small) study whose main contribution is in the approach (i.e. Bayesian analysis on colonization and extirpation relative to random), rather than in the finding (i.e. species with low N scores are more likely to be gained and lost, species with higher temp scores are preferentially gained and those with lower temp scores are preferentially lost). The approach is very novel and powerful, and I look forward to seeing it replicated with a larger dataset. I have no major objections to the paper, but would have liked to see more sites and a wider geographic gradient to see how robust the results are outside of this small eco-type (i.e. Swiss mountain meadows).

I had no major comments, and consider the above comments as moderate, to improve the quality and clarity of the paper. Minor comments are below:

Line 45: Probably good to give the formal definition of the CL here from Nilsson et al. (1988) or Bobbink et al. (2010).

Line 76: “say’ is too colloquial.

Line 78: Include some discussion of fire (i.e. it does not happen?) since that is also a very important factor in grasslands.

Table 1: Seems odd that such small changes can be significant. For example, light went from 3.56 to 3.55, and you found a 93% chance that this is a real trend. That seems a little questionable.

Line 156-167: This analytical method is very difficult to follow, please clarify.

Line 187: Clarify that “replicated surveys” are where you had different botanists measure the sites, replicates can have multiple meanings here (e.g. replicate plots measured by the same person). These are “validation plots.”

Line 201-206: Very interesting results. New colonizers are adapted to low N and are different from the extant community, species lost are also adapted to low N conditions. Interesting that you were able to detect this!

Line 235-236: That you did not detect a shift is not surprising if these are perennial species (are they?) Some of these grasses could live for years and thus a shift may not be expected when the reductions in N deposition are this small.

Line 254-256: This is a bit of an overstatement, please water down a bit.

Line 257: Compare the reduction nin N dep (-2.7 kg ha-1 yr-1) with the interannual variation, this is probably within the interannual variation and thus is not a large enough reduction to detect more definitive changes. That being said, you did detect some, which again bespeaks the strong analytical approach.

Line 266-267: These results sort of makes sense, N dep is coming down (favoring those with lower N scores), but it's still high relative to historical levels (so you're still going to lose species adapted to low N conditions).

Line 289-290: It’s unclear to me which “space-for-time” substitution you’re comparing your results to. Is it Rowe et al. (2017)? Whatever it is, please cite it in this sentence, and put your number next to theirs for easy comparison.

Line 293: Please identify what the “same model as in Roth et al. (2017)” is.

Line 309: Please re-word, “re-transferring mountain hay meadows,” I don’t know what you mean by re-transferring here.

Line 311-312: “However…instead of species turnover.” This is kind of an unsubstantiated claim, either reword to make it a hypothesis (i.e. water it down), or back it up.

---

## Round 0.2 · accepted · Accept

The authors have responded adequately to the reviewers' comments and suggestions and I find the manuscript improved over the first submitted version. This paper will make a good addition to our understanding of how plant communities are responding to decreases in atmospheric N deposition as the climate simultaneously changes.